# Impact of Six Extraction Methods on Molecular Composition and Antioxidant Activity of Polysaccharides from Young Hulless Barley Leaves

**DOI:** 10.3390/foods12183381

**Published:** 2023-09-09

**Authors:** Mingming Wang, Chuangchuang Zhang, Yuting Xu, Mengting Ma, Tianming Yao, Zhongquan Sui

**Affiliations:** 1Department of Food Science & Technology, School of Agriculture and Biology, Shanghai Jiao Tong University, Shanghai 200240, China; 3030336726@sjtu.edu.cn (M.W.); cczhang@sjtu.edu.cn (C.Z.); nov16xyt@sjtu.edu.cn (Y.X.); mmt9497@sjtu.edu.cn (M.M.); 2Department of Food Science, Whistler Center for Carbohydrate Research, Purdue University, 745 Agriculture Mall Drive, West Lafayette, IN 47907, USA

**Keywords:** polysaccharide, fiber extraction, hulless barley, monosaccharide composition, antioxidant activity

## Abstract

Young hulless barley leaves are gaining recognition for potential health benefits, and the method of extracting polysaccharides from them is critical for potential food industry applications. This study delves into a comparative analysis of six distinct fiber extraction techniques: hot water extraction; high-pressure steam extraction; alkaline extraction; xylanase extraction; cellulase extraction; and combined xylanase and cellulase extraction. This analysis included a thorough comparison of polysaccharide–monosaccharide composition, structural properties, antioxidant activities (DPPH, ABTS, and FRAP), and rheological properties among fibers extracted using these methods. The results underscore that the combined enzymatic extraction method yielded the highest extraction yield (22.63%), while the rest of the methods yielded reasonable yields (~20%), except for hot water extraction (4.11%). Monosaccharide composition exhibited divergence across methods; alkaline extraction yielded a high abundance of xylose residues, whereas the three enzymatic methods demonstrated elevated galactose components. The extracted crude polysaccharides exhibited relatively low molecular weights, ranging from 5.919 × 10^4^ Da to 3.773 × 10^5^ Da across different extraction methods. Regarding antioxidant activities, alkaline extraction yielded the highest value in the ABTS assay, whereas enzymatically extracted polysaccharides, despite higher yield, demonstrated lower antioxidant capacity. In addition, enzymatically extracted polysaccharides exerted stronger shear thinning behavior and higher initial viscosity.

## 1. Introduction

Hulless barley (*Hordeum vulgare* L. var. nudum Hook. f) emerges as a remarkable cereal crop, well-known for its multifarious nutritional advantages and positive impacts on health. The native highland hulless barley species cultivating on the Qinghai–Tibet Plateau (China) plays a pivotal role in local life, which serves as an irreplaceable ingredient for a wide range of regional foods, including beverages, bakery items, and dietary supplements [1]. An essential attribute that sets hulless barley apart is its robust nutritional profile, enriched with dietary fiber and antioxidants [2]. Additionally, the fiber-rich feature of barley underscores the potential to contribute significantly to dietary fiber intake, which has been linked to a range of health benefits, including improved digestion and reduced risk of chronic diseases [3,4]. Interestingly, although traditional barley consumption witnessed a decline globally, a renewed interest in barley leaves emerged due to its exceptional nutritional components. Hulless barley grass (refers to the young leaves and stems) has garnered special attention for its popular use as a green-colored juice made with young leaf powders in Asia and the value of its functional elements that have been associated with a number of health advantages [5]. The functional extracts from young barley leaves have been reported to promote human lipid metabolism [6] and blood glucose regulation [5] as well as host antidepressant [7], antiaging [8], antioxidant [9,10], and antidiabetic effects [10]. These multi-faceted benefits underline the promising potential of young barley leaves as a functional approach to health and well-being. The daily consumption of young green hulless barley powders by Tibetans and people in Southwest China provinces has been linked to lower cancer mortality and extended longevity, even amidst challenging living conditions in these regions [11]. In addition, the potentially impactful bioactive compounds in young barley leaf juice include phenolic molecules, alkaloids, flavonoids, and chlorophyll, which have been extensively studied [9,12].

Although polysaccharides constitute a significant portion of barley leaf powders and offer valuable health benefits, they have not been thoroughly investigated with the same depth of evaluation as the phytochemical molecules (e.g., polyphenol, flavone) in barley leaves. A significant challenge lies in the establishment of a dependable and efficient extraction method for the functional polysaccharides within insoluble leaf powders. Unlike small molecules that can be readily dispersed using solvents, functional polysaccharides with higher molecular weight are naturally entrapped within plant cell wall matrices, intertwined with cellulose and lignans, resulting in a difficult obstacle to isolate and investigate their functions [13]. Moreover, fiber solubility poses another issue, which is closely linked to both bio-accessibility upon consumption and suitability for various application forms in the food industry. Hence, the extraction methods play a key role in assessing the function of polysaccharides from hulless barley leaves, which is also of importance for downstream food applications since the extraction yield and the content of extracts from different methods can be distinct.

The traditional polysaccharide extraction methods for cereal plants include hot water extraction and alkaline extraction. Both established approaches have been widely used to extract soluble polysaccharides from diverse cereal sources [14]. However, concerns arise from their low yield and extensive chemical usage, which limit their application in food products. Recently, novel extraction techniques have been developed and reported, including high-pressure steam [15], solid-state fermentation by *Rhizopus* sp. [16], and enzymatic extraction using xylanase or cellulase [14]. High-pressure steam employs heat energy and elevated pressure to break internal covalent bonds, reducing large molecules to smaller sizes and enhancing water solubility. Enzymatic extraction using recombinant xylanase or cellulase offers benefits like lower reaction temperature, reduced labor, energy savings, and high enzyme specificity.

Although each of these methods has its own strengths and limitations and has been individually investigated for diverse cereal fiber extractions, a comprehensive comparison among them, specifically regarding the fiber extraction from green hulless barley powders, is currently absent in the literature. In this study, we aimed to assess and compare six distinct polysaccharide extraction methods for young hulless barley leaves. These methods can be categorized into physical (hot water and high-pressure steam), chemical (alkaline), and enzymatic (xylanase, cellulase, and a combination of both) approaches. A comparative analysis was conducted to assess the extraction yield, molecular composition, and structural information of the extracts, viscosity properties, and their potential antioxidant activities.

## 2. Materials and Methods

### 2.1. Materials

The hulless barley seeds were obtained from Qinghai Xinning Biological Technology Company (Xining, China). Seeds were thoroughly soaked in water and placed onto a seedling tray filled with a thin layer (0.5 cm) of nutrition-rich soil. After a few days of cultivation, the seeds germinated, and the young green leaves were harvested on the fifteenth day. The harvested leaves included the portion that was exposed above the ground. Leaf samples were washed thoroughly with deionized (DI) water to remove any soil and contaminants. After cleaning, leaves were freeze-dried for 24 h and then ground to green powders using a grinder and passed through a sieve (40 mesh). The protein, lipid, soluble and insoluble dietary fiber, and dietary fiber content of the young leaf powders (dry basis) were 28.5%, 12.9%, 1.6%, and 47.4%, respectively [17]. The α-amylase (EC 232-588-1, 30 U/mg), papain (EC 232-627-2, 10 U/mg), xylanase (EC 253-439-7, 2500 U/g), and cellulase (EC 232-734-4, 5000 U/g) were purchased from Sigma-Aldrich Co. Ltd. (Saint Louis, MO, USA). All other chemicals used in this study were of chemical grade and from Sinopharm Chemical Reagent Co., Ltd. (Shanghai, China).

### 2.2. Starch and Protein Removal from Young Hulless Barley Leaf Powders

Dried hulless barley leaf powders (20 g, dry basis) were dispensed in 500 mL of DI water and boiled for 10 min. After cooling down to 60 °C, 5000 U α-amylase and 20,000 U papain were added to degrade starches and proteins for 1 h. Next, 6000 U amyloglucosidase was added into the slurry at pH 4.5, incubating for another 1 h at 60 °C. The digested powders were then heated to 100 °C to inactivate enzymes and centrifuged at 1400× *g* for 15 min. The pellets were washed with DI water and air dried at 37 °C. The obtained insoluble fiber, free from starch and protein, was ground, passed through a 40-mesh sieve, and prepared for the downstream extraction. 

### 2.3. Extraction of Hulless Barley Leaf Polysaccharide through Different Methods

Six distinct methods were adopted in this study to facilitate a thorough comparison of their extraction yield, the compositions of extracted fibers, the structural property, the rheological property, and their potential antioxidant capacity. The extraction yield is determined as the ratio of the dry weight of extracted polysaccharide powder and the dry weight of initial hulless barley leaf powders. The following acronyms are used for different samples: HW for hot-water extraction; HPS for high-pressure steam extraction; AE for alkaline extraction; XE for xylanase extraction; CE for cellulase extraction; and XCE for combined xylanase–cellulase extraction.

#### 2.3.1. Hot-Water Extraction

Hot-water extraction was conducted according to a well-established protocol [18] with the optimal temperature of 80 °C and 2 h for leaf samples. Briefly, a quantity of 2 g dried insoluble fiber powders (from Section 2.2) was combined with 50 mL of deionized water (1:25 *w*/*v*) and subjected to incubation at 80 °C for a duration of 2 h. The choice of solid–liquid ratio (1:25) was according to a previous parametric extraction study [17], and the ratio was consistently applied across all extraction methods in this study. After the mixture was cooled to room temperature, vacuum filtration was conducted using a Buchner funnel. The resultant filtrate was carefully collected and then slowly introduced to four volumes of 95% ethanol. This mixture was allowed to precipitate overnight at 4 °C. The slurry was centrifuged at 1400× *g* for 20 min, and the precipitates were rehydrated to a volume of 50 mL and subsequently processed through rotary evaporation to a final volume of 5 mL. The obtained sample was then freeze-dried and preserved for further analysis.

#### 2.3.2. High-Pressure Steam Extraction

Dried insoluble fiber powders from Section 2.2 (2 g) were placed on a petri dish (120 mm diameter) covered with the lid and then autoclaved with a liquid cycle (235 KPa, 121 °C) for 30 min. Following a cooling step, the fiber powders were dispensed in DI water (50 mL) and further incubated at 80 °C for 2 h. The yielding slurry was vacuum filtrated, and the filtrate was collected. The alcohol precipitation and rotary evaporation steps were conducted as per the description in Section 2.3.1. 

#### 2.3.3. Alkaline Extraction

Dried insoluble fiber powders from Section 2.2 (2 g) were dispensed in NaOH solution (0.5 M, 50 mL) and incubated at 80 °C for 2 h. The sample slurry was cooled down to room temperature, and then vacuum-filtrated. The subsequent alcohol precipitation and rotary evaporation steps were the same as described in Section 2.3.1.

#### 2.3.4. Extraction with Xylanase

A quantity of 2 g of dried insoluble fiber powders (from Section 2.2) was mixed with 50 mL of DI water, and the pH was adjusted to 5.0. Xylanase (1600 U) was mixed into samples and incubated at 50 °C for 3 h. The subsequent filtration, alcohol precipitation, and rotary evaporation steps were the same as described in Section 2.3.1.

#### 2.3.5. Extraction with Cellulase 

Two grams of dried insoluble fiber powders (from Section 2.2) was mixed with 50 mL DI water. The pH of the mixture was subsequently adjusted to 5.0. Then, 800 U cellulase was added and mixed with samples at 50 °C for 3 h. The following filtration, alcohol precipitation, and rotary evaporation steps were the same as described in Section 2.3.1.

#### 2.3.6. Combined Extraction with Both Xylanase and Cellulase

Dried insoluble fiber powders from Section 2.2 (2 g) were dispensed in 50 mL DI water, and the pH was adjusted to 5.0. Both xylanase (1600 U) and cellulase (800 U) were blended with samples and incubated at 50 °C for 3 h. The subsequent steps involving filtration, alcohol precipitation, and rotary evaporation mirrored the procedures detailed in Section 2.3.1.

### 2.4. Monosaccharide Composition of Different Extracted Fibers

Each extracted soluble fiber sample (2 mg) was subjected to treatment with 3 mL of 2 M trifluoroacetic acid, maintained at a temperature of 110 °C for 4 h. The hydrolyzed samples were air dried, followed by a wash with 1 mL of methanol, repeated three times to ensure complete dehydration. The fully dried samples were adjusted to a final volume of 50 mL with DI water and passed through a 0.22 μm syringe filter. A volume of 25 μL from each sample was injected into a Dionex™ ICS-5000+ high-pressure ion chromatography system (HPIC, Thermo Fisher Scientific Inc., Waltham, MA, USA), equipped with a CarboPac™ PA20 column (3 × 150 mm) and a pulsed amperometric detector (PAD). Elution of mobile phases (DI water, 1 M CH_3_COONa, and 250 mM NaOH) was performed using gradient mode at a flow rate of 0.5 mL/min. Monosaccharide standards were used to identify the sample peaks, including fucose (Fuc), rhamnose (Rha), arabinose (Ara), galactose (Gal), glucose (Glc), xylose (Xyl), mannose (Man), fructose (Fru), galacturonic acid (GalA), and glucuronic acid (GlcA).

### 2.5. Molecular Weight Determination

The molecular weight (Mw) of each fiber obtained via different extraction methods was measured on a high-performance size exclusion chromatography (HPSEC) system (1260 Infinity II, Agilent Technologies, Inc., Santa Clara, CA, USA) equipped with a series connection of an SB-805 HQ (8.0 × 300 mm, 13 μm) column and an SB-803 HQ (8.0 × 300 mm, 6 μm) column (OHpak, Showa Denko K.K., Osaka, Japan). An Optilab T-rEX refractive index (RI) detector and a DAWN HELEOS-II multi-angle laser light scattering (MALLS) detector were used to record the chromatogram and estimate the Mw of samples. Samples were dissolved into mobile phase (0.1 M NaNO_3_ with 0.02% *w*/*v* NaN_3_, 60 °C) to a concentration of 1% (*w*/*v*) and 0.22 μm syringe filtered before injection.

### 2.6. Fourier Transform Infrared (FTIR) Analysis

The FTIR spectra of polysaccharides extracted from young hulless barley leaves were acquired following the procedure outlined by Li et al. [19]. The Fourier transform infrared spectrophotometer (Nicolet IS10, Thermo Fisher Scientific Inc., Waltham, MA, USA) equipped with a Universal ATR platform (Thermo Fisher Scientific Inc., Waltham, MA, USA) was used. In brief, 2 mg of dried polysaccharide samples was directly loaded onto the diamond surface of the conical accessory plate. Spectral data were collected using a mid-range beam with a wavelength of 4000 to 500 cm^−1^ and a resolution of 4 cm^−1^. Each measurement was conducted in duplicate. Data were analyzed using OMNIC 8.2 (Thermo Nicolet Corporation, Madison, WI, USA).

### 2.7. Bioactivity Analyses

DPPH (1,1-diphenyl-2-picryl-hydrazyl) radical scavenging test was performed according to Guo et al. [20]. In brief, hulless barley young leaf fibers (10 mg, dry basis) were dissolved in 3 mL deionized (DI) water to obtain a solution. Subsequently, a sample solution of 100 μL was mixed with 100 μL of a DPPH ethanol solution (0.1 mM) within a 96-well plate. The DPPH radical solution’s color changed from violet to yellow after a 30-minute reaction in the dark at room temperature. Absorbance was measured at λ = 517 nm. The 6-hydroxy-2,5,7,8-tetramethylchroman-2-carboxylic acid (Trolox) was used as the assay standard, with a concentration gradient (0, 50, 100, 200, 300, 400 μM). The DPPH radical scavenging ability of the samples was determined as Trolox equivalents value (μM TE/g).

The ferric-reducing antioxidant power (FRAP) assay follows the method outlined by Benzie and Strain [21]. Sample aliquots (3.33 mg/mL) were mixed with the FRAP reagent and measured immediately at wavelength 593 nm. Standard solutions of FeSO_4_ with variant concentrations (100, 200, 400, 600, 800, and 1000 μM/L) were used for calibration. Outcomes were expressed as the FeSO_4_ equivalents of antioxidant capacity (μM FeSO_4_/g).

The 2,2′-azino-bis(3-ethylbenzothiazoline-6-sulfonic acid) (ABTS) free radical-scavenging activity of each extracted sample was determined according to the method described by Arts et al. [22]. To create the ABTS stock solution (7.4 mM), potassium persulfate (2.6 mM) was combined in equal proportions, followed by an overnight (16 h) incubation in darkness before usage. Leaf polysaccharide solutions (3.33 mg/mL) were then mixed with the ABTS solution at a 1:4 ratio to 200 μL and incubated in the dark for 10 min prior to measurement at a wavelength of 734 nm. Calibration curves were generated using Trolox solutions as standards, following a similar approach as for the DPPH method. The scavenging capacity of the samples against ABTS radicals was also quantified in terms of Trolox equivalents of antioxidant capacity (μM TE/g).

### 2.8. Rheological Property

Fiber solutions were prepared at the concentration of 1% (*w*/*v*). The viscosity of each sample solution was determined using a rheometer (ARES-G2, TA instruments, Wilmington, DE, USA) with a 40 mm 2 cone geometry. The sample measurements were conducted at a 1000 μm gap and with the shear rate ranging from 0.1 to 100 s^−1^ at 25 °C. The results were fitted to the following power law model:

σ = K∙γ^n^, in which σ is the shear stress (Pa); K is the consistency coefficient (Pa∙s^n^); n is the flow behavior index, and γ is the shear rate (s^−1^).

### 2.9. Statistical Analysis

The data were derived from three technical triplicates and are presented as mean values ± standard error. Statistical comparisons were conducted using analysis of variance (ANOVA). To assess group differences, Fisher’s least significant difference (LSD) tests with Bonferroni adjustment were used (*p* < 0.05). All these statistical analyses were carried out using SPSS 17.0 (SPSS Inc., Chicago, IL, USA). Pearson correlation coefficients analysis was performed using R 4.2.2 (R Core Team, 2022, R Foundation for Statistical Computing, Vienna, Austria) along with the Hmisc package.

## 3. Results and Discussion

### 3.1. Extraction Yield

Figure 1 illustrates the extraction yield achieved by the six selected methods. Generally, the enzymatic extraction methods demonstrated a notably higher overall extraction yield when compared to chemical and physical extraction approaches. Notably, XCE extraction achieved the highest yield of 22.63 ± 0.22%, which was significantly (*p* < 0.05) higher than that of the other methods. In contrast, the HW method merely reached a yield of 4.11 ± 0.38%, which was 5.5 times lower than that of XCE. However, the HPS method, another physical approach, achieved a superior yield compared to HW. The HPS yield level aligned with that of chemical methods and single enzymatic methods. This improved yield can be attributed to the robust physical disruption of the young hulless barley leaves, which occurs during the release of high-pressure steam in the process, leading to an explosive depolymerization of the leaf tissue [23]. As a result, the cell wall structure opens up, and hemicelluloses are exposed, facilitating their straightforward extraction. Although the high pressure and elevated temperature associated with steam treatment are energy-intensive for the extraction, the steam explosion process is recognized for its environmentally friendly nature and its capacity to yield effective outcomes.

The chemical extraction method (i.e., AE) attained a lower extraction yield (19.2%) than XCE. The utilization of alkaline chemicals for fiber extraction is a well-established technique that has been extensively studied, with various changeable parameters such as alkaline agents and their concentrations. AE operates on the mechanism that hydroxyl ions break hydrogen and covalent bonds between hemicellulose and cell wall cellulose matrices. In addition, ester bonds that cross-link several hemicellulose chains can also be disrupted, leading to the solubilization of hemicellulose fibers [24]. Although there is inadequate information regarding the extraction yield of fibers from barley leaves, studies on alkaline extraction from fiber-rich portions of barley, such as barley husks, reported yields ranging from 25.0% to 31.1% [25]. Given that sprouted cereal leaves were found to contain slightly lower fiber content than grains [26], the achieved extraction yield with the AE method of approximately 19.2% is reasonable. 

Enzymatic degradation is an alternative and more efficient method for fiber extraction, which emerged a few decades ago, initially using carbohydrate metabolism enzymes sourced from bacteria [27]. Both the XE and CE methods achieved extraction yields at a comparable level, whereas the combined XCE method exhibited a significantly high yield. This underscores the synergistic effect resulting from the concurrent action of two enzymes. It has been reported that the combined enzymatic digestion can yield up to a 60% increase in extracted fibers [14]. In our study, the XCE method yielded a 20% increase in comparison to single enzyme treatments, prompting consideration for further enhancement in extraction time to optimize the yield. The application of endo-β-(1,4)-xylanases is a relatively recent development in solubilizing fibers in cereal-derived by-products [28]. These enzymes efficiently reduce the molecular size of xylan-backboned fibers, resulting in smaller fiber sizes, which, in turn, enhance solubility and facilitate extraction. On the other hand, cellulase primarily degrades the insoluble cellulose portion, aiding in the release of soluble fibers, but it cannot improve the intrinsic solubility of extractable fibers. As a result, molecular structures and fiber compositions from xylanase and cellulase treatments should differ. The CE method is expected to yield longer fiber chains, while the XE method may yield fewer substituted xylan fibers. This encourages further investigation into the extracted fiber compositions and molecular size distributions in the following sections.

### 3.2. Monosaccharide Composition of Hulless Barley Leaf Polysaccharides Extracted Using Different Methods

The composition of neutral sugars and uronic acids in each fiber obtained through various methods has been summarized in Table 1. Galactose (24.54–47.84 mol%) and arabinose (16.20–20.94 mol%) were consistently detected as common constituents in hulless barley leaf fibers across all extraction methods, with fucose, rhamnose, mannose, and two uronic acids present as minor sugar components. However, it was evident that different extraction methods yielded varying predominant monosaccharide units in each fiber. For instance, the content of glucose units ranged from 7.06 to 34.00 mol%, with the highest content observed in HW, followed by CE, AE, HPS, XCE, and XE. Similarly, the xylose unit content (6.60–36.24 mol%) exhibited a wide dispersion. AE produced the highest xylose content, followed by HPS, XE, HW, CE, and XCE. When comparing the differences between physical, chemical, and enzymatic extraction methods, the three enzymatic methods demonstrated internal consistency, displaying notably higher galactose and arabinose content in comparison to the other methods. AE, the chemical modification method, exhibited higher xylose content but lower galactose content, suggesting a higher content of xylan-based fibers in comparison to the other methods. The two physical extraction methods displayed distinct differences: HW contained a higher amount of glucose, while HPS exhibited elevated levels of both galactose and xylose, indicating a more heterogeneous composition of fiber components.

In the literature, the commonly extractable dietary fibers from cereal cell wall’s non-starch polysaccharides have mainly fallen into the categories of arabinoxylans, β-glucans, arabinogalactans [29], as well as pectin [30]. The presence of arabinose, galactose, galacturonic acid, glucose, and xylose units in this study aligns well with the constituents of the aforementioned potential polysaccharide molecules. However, notable variation exists in the abundance of these polysaccharide molecules among the extracts obtained from different methods. Notably, HW appears to contain two–three times more β-glucans than all other methods, as evidenced by its high glucose content. Since a starch removal pre-treatment was employed during the sample preparation, the observed glucose content here should be readily attributed to the non-starch glucan portion. β-glucan is known for its high water solubility, attributed to its (1 → 3)-linkages, which gives rise to an irregular linear structure that inhibits intermolecular association [31]. Conversely, other polysaccharides like arabinoxylan are more prone to crosslinking with multiple molecules, thereby impeding dissolution in hot water. In a study by Ahmed et al. [32], β-glucan was extracted from barley using four methods: hot water; acid; alkaline; and a combination of amylase and protease. They reported that hot water extraction yielded the highest amount of β-glucan, suggesting that it was the most suitable approach for extracting β-glucan to minimize impurities. This observation matches our results, even though the HW method exhibited the lowest extraction efficiency. 

Evidently, the AE method appeared to be favorable for the extraction of arabinoxylan, displaying an over five-fold increase in xylose units in mole percent when compared to high-yield enzymatic methods like XCE. Due to the exclusive presence of xylose units within the structure of arabinoxylan molecules, the high abundance of xylose units can be confidently attributed to the content of arabinoxylan polymers in the fiber mixture. One plausible mechanism for this high arabinoxylan finding is the existence of esterification bonds among cell wall arabinoxylan chains or between arabinoxylan and cellulose polymers. Such cross-linking might be facilitated by di-ferulic acids and p-coumaric acids [33], giving rise to covalent bonds that resist disruption by conventional physical methods or common digestive enzymes, such as amylase, cellulase, or xylanase [34]. In order to break down these esterification bonds, the implementation of either a specialized enzyme known as carbohydrate esterase or the use of a strong alkaline condition, which induces hydroxyl ion repulsion, is required. Given that our enzymatic extraction methods did not involve the use of any esterase, only the AE method exclusively appeared to be a suitable approach for yielding fibers rich in arabinoxylan content.

Among the three enzymatic extraction methods, there is a notably higher presence of galactose and rhamnose in the extracted fibers. This occurrence could potentially be linked to the composition of pectic polymers. Pectin, being a complex polysaccharide, contains four distinct polymeric regions: homogalacturonan; xylogalacturonan; rhamnogalacturonan type I (RG-I); and rhamnogalacturonan type II (RG-II). Among them, RG-I is the most intricately branched polymer, characterized by a substantial proportion of long side chains, including arabinan, galactan, and arabinogalactan [35]. Kim et al. [36] found that enzymatic digested young barley leaves yielded a fiber that displayed a substantial content of xylose and galactose units, closely matching our findings. Furthermore, their subsequent experiment involving pectinase digestion on these fibers effectively broke down the galactose-rich portion, which strongly suggested that the enzymatically extracted fiber from barley leaves had a complex composition of arabinoxylan- and RG-I-rich polysaccharides [30]. The effectiveness of the pectinase treatment underscores the correlation between the high galactose units present in the leaf extracts and the pectic polymer structure. Therefore, our fibers obtained via CE and XCE methods in our study are also expected to contain a considerable abundance of RG-I pectin, hypothetically constituting more than 60 mol% (e.g., rhamnose, galactose, and galacturonic acid) of the overall polysaccharide contents.

### 3.3. Molecular Weight Distribution of Extracted Hulless Barley Leaf Polysaccharides

Table 2 presents a summary of the weight average molecular weight (Mw), number average molecular weight (Mn), and polydispersity index (PDI) values for the highest molecular size peaks obtained from the six extracted fibers. Each polysaccharide extract consisted of a mixture of various fiber molecules. The size exclusion chromatography (SEC) chromatograms exhibited broad and wide peaks, with Mw ranging from 5.919 × 10^4^ Da to 3.773 × 10^5^ Da. Interestingly, enzymatic extraction yielded polysaccharides with higher Mw compared to the alkaline method and the physical extraction methods. This finding may seem counterintuitive, given that enzymatic digestion typically leads to glycosidic linkage cleavage and a subsequent reduction in molecular weight, particularly with xylanase that targets the arabinoxylan backbone. However, a study by Chen et al. [37] reported an increase in Mw through enzymatic extraction compared to alkaline and hot water methods from fresh ginger leaves. The reduced Mw observed in hot water or alkaline methods could be attributed to the influence of high temperatures, extreme pH conditions, and prolonged extraction durations [38]. 

When examining the overall Mw of the polysaccharides extracted from hulless barley leaves, the size distribution appeared to be relatively low in comparison to fibers extracted from cereal seeds, which have been reported to cross the megadalton range. These differences could be attributed to the unique biocomposition of leaves in contrast to seeds, with barley leaf cells containing more abundant pectic compounds [30]. Our observation aligns with a study by Xu et al. [39], which stated that the polysaccharides derived from hulless barley grass exhibited an Mw of 2.99 × 10^4^ Da. 

### 3.4. FTIR Spectrum Analysis

As shown in Figure 2, the FTIR spectra obtained via six different extraction methods provide additional structural insights into the barley leaf extracts. Generally, the peaks observed in the samples are characteristic bands related to the structure of polysaccharides within the carbohydrate fingerprint region (800–1200 cm^−1^) [40]. These features are consistent across various samples and include the 1122 cm^−1^ intermediate O–H shear vibration band and the 1093 cm^−1^ glycosyl linkage (C–O–C) stretching vibration band. A broad O–H stretching peak at 3320 cm^−1^ and a distinct bump at 2940 cm^−1^ for C–H stretching were observed in all samples, suggesting the possible presence of molecular structures containing carboxyl and hydroxyl groups. Notably, this broad band at 3000–3600 cm^−1^ is commonly presented in various polysaccharide samples [41]. The band at 1633 cm^−1^ possibly corresponds to the stretching vibration of a C=C bond or a C=O bond related to carbohydrate structures. The presence of a peak at 1432 cm^−1^ indicates the potential carboxylate doublet stretching, which has been previously associated with pectic polysaccharides [40]. Interestingly, the fiber extracted via the AE method exhibited an additional peak in the fingerprint region at 1047 cm^−1^, which could be indicative of a structure rich in hemicellulose, potentially arabinoxylan [42]. This finding is consistent with the higher xylose content observed in the AE method in the monosaccharide analysis. Among the enzymatic extraction methods, the XE and CE samples displayed similar IR spectra, while XCE demonstrated a distinct pattern with additional bands at 993 cm^−1^. This band is associated with C–OH bending and C–O–C glycosidic bond vibration [43], indicating a potentially unique carbohydrate configuration. The bands below 800 cm^−1^ are known as the skeletal region for polysaccharides; however, limited information is available on this region, and it is not extensively covered in the literature. It is also noticeable that there are no excessive peaks (besides the 1633 cm^−1^) within the phenolic group fingerprint region (1500–1800 cm^−1^ range), meaning that all six extraction methods are good for polysaccharide extraction but not for phenolic compounds. The FITR spectra supply additional molecular structure information of polysaccharides and underscore the structural diversity of the extracted polysaccharides.

### 3.5. Radical Scavenging Bioactivities of Extracted Hulless Barley Leaf Fibers

The investigation into the antioxidant activity across a diverse range of food compounds has garnered increasing research interest, with the potential to mitigate oxidative stress and ameliorate disease states within the human body [44]. In the recent field of antioxidant assays, the focus has shifted from isolated single compounds to the consideration of the entire group of mixed compounds within a sample. As a result, the extensive diversity and spectrum of antioxidant compounds found in actual foods requires the use of various assays grounded in different chemical mechanisms. This approach facilitates a broad understanding and a more comprehensive, synergistic interpretation of the intricate bioactivity of health-beneficial food products, which holds particular promise in exploring the potential health advantages against oxidative stress-related disorders [45]. A straightforward mechanism explaining the antioxidant capacity of polysaccharides remains elusive since this challenge derives from the complicated polysaccharide structures and chemical composition variability. There are increasing investigations to reveal the potential novel antioxidant properties of polysaccharides. For example, pectic polysaccharides themselves can act as free radical scavengers with rich hydroxyl groups, carboxyl groups, and other functional groups that can neutralize free radicals by donating electrons [46]. In addition, natural arabinoxylans often feature feruloylation, supporting them with antioxidant potential through these covalently bound functional groups. Extensive reports also exist regarding the potential antioxidant behavior of β-glucan, which is based on both the direct binding to reactive species and the activation of in vivo antioxidant systems [47]. 

Figure 3 illustrates the in vitro antioxidant capacity of the six hulless barley leaf fibers measured with three distinct assays: DPPH (Figure 3A); ABTS (Figure 3B); and FRAP (Figure 3C). The DPPH assay, renowned for its rapid and precise measurement of antioxidant activity, has been widely applied in assessing the potential of various plant extracts [48,49]. In our study, diverse fiber extraction methods exhibited variances in the DPPH radical scavenging assay. Notably, fibers extracted with the HPS method demonstrated the highest DPPH radical scavenging ability, followed by AE and three enzymatic methods followed in sequence. The HW fiber displayed the lowest antioxidant capacity. These findings suggested that although the combined enzymatic extraction (XCE) yielded higher polysaccharide content, the corresponding bioactivity might not be commensurately elevated.

However, a distinct trend emerged in the ABTS assay, where AE exhibited the highest antioxidant potency, followed by two physical methods (HPS and HW) and then the enzymatic methods (CE, XE, XCE). Interestingly, despite significant variations in the DPPH results, both assays indicated that the enzymatic-based fiber extraction methods did not yield substantial antioxidant activity. Although DPPH is more generally used for antioxidant determination, our results suggest that the ABTS assay may better reflect the antioxidant contents in the barley leaf extracts, which can be attributed to the leaf polysaccharides being water-soluble but insoluble in ethanol. Both the DPPH and ABTS assays operate through the sequential proton loss electron transfer (SPLET) mechanism. This mechanism involves the initial loss of a proton from the antioxidant, followed by anion transfer to the radical, which subsequently reacts with the proton [50]. However, the reaction is sensitively influenced by the proton affinity, electron transfer enthalpy, pH, and solvent types. One noticeable difference between the DPPH and ABTS tests lies in their solvent preferences. DPPH favors an alcoholic solvent, whereas ABTS assay performs optimally in a water-based solvent [51]. In addition, some studies have demonstrated that the ABTS assay is better suited for assessing antioxidant activity in food matrices, displaying lower variability compared to DPPH since DPPH is particularly adept at measuring small molecule compounds that dissolve in alcoholic solutions [45,51]. These findings matched our observation that the ABTS test exhibited lower deviation and rendered more consistent results than the DPPH assay. 

The FRAP assay, another antioxidant test, presents an almost opposing viewpoint regarding the antioxidant activity of the fiber, showing relatively robust capacity for fibers obtained from enzymatic extraction methods (CE and XE), similar to that observed for the physical methods (HPS and HW) in this test. Conversely, AE and XCE displayed lower Fe^3+^ ion reduction capacity. The FRAP assay diverges from the other assays as it does not involve free radicals. It monitors the reduction in ferric iron (Fe^3+^) to ferrous iron (Fe^2+^). Therefore, this test provided a distinct perspective by assessing the sample’s reducing power rather than its free radical scavenging ability. Several other studies have also reported differing trends between ABTS and FRAP assays when evaluating plant products. Ou et al. [52] noted the absence of a correlation between FRAP and Trolox equivalent-based assays across the majority of their 927 freeze-dried vegetable extracts, suggesting the inherent inconsistency among various antioxidant assays. Furthermore, in a statistical correlation study, the FRAP result was found to be positively linked with the total phenolic compounds quantified using Folin reagents rather than with ABTS or DPPH measurements [53]. Therefore, the lower FRAP value but higher ABTS value observed for the AE fiber simply indicates strong radical scavenging activity coupled with low reducing power. 

As shown in Table 3, only the ABTS assay exhibited a strong correlation with the compositional makeup of hulless barley leaf polysaccharides. Given that ABTS is a suitable assay for representing the bioactivity of samples, as discussed earlier, we can confidently link the presence of xylose sugar units in leaf fibers to a more potent antioxidant activity, substantiated by a high coefficient score of 0.92 (*p* < 0.01). It indicated that the high abundance of arabinoxylan molecules, as anticipated in AE and HPS fibers, underpinned the enhanced radical scavenging capacity. However, the significant negative coefficient between ABTS and galactose (or rhamnose) implied that the portion of pectic polysaccharides in extracts might not significantly contribute to the antioxidant properties. The strong correlation between arabinoxylan and robust antioxidant activity can be attributed to the ester-linked phenolic groups present in arabinoxylan [33]. The residual free polyphenols or bound phenolic molecules on the polysaccharides might also contribute to the antioxidant function to a certain extent. However, due to the relatively limited quantity of phenolics within polysaccharide extracts and the complex nature of the fibers, characterizing every compound and elucidating their individual contributions is a complex endeavor. Hence, for a comprehensive understanding, further phytochemical assays and a detailed compositional analysis of compounds beyond polysaccharides will be required for a future study.

### 3.6. Rheological Property

Figure 4 illustrates the apparent viscosity as a function of shear rate for six hulless barley fibers extracted using different methods. The apparent viscosity of polysaccharides is a crucial functional property closely related to their potential applications in the food industry. The apparent viscosity profiles of the six polysaccharide samples exhibited notably different patterns, particularly at low shear rates. At a shear rate of 1/s, the highest apparent viscosity was observed in the fiber extracted with XCE (0.0619 Pa·S), followed by CE (0.0143 Pa·S), XE (0.0057 Pa·S), AE (0.0045 Pa·S), HW (0.0043 Pa·S), and HPS (0.0039 Pa·S). Additionally, we employed the power law model to describe the flow behaviors, and the associated parameters are summarized in the inset table within Figure 4. The flow consistency index (K) fell within the range from 0.0023 to 0.4831, which aligned with the apparent viscosity values observed at lower shear rates. All polysaccharides exhibited non-Newtonian fluid characteristics, characterized by a flow behavior index (n) that is less than 1, indicative of a pseudo-plastic behavior. There was a significant difference in the n index, with XCE and the other two enzymatic extraction methods displaying lower n values than the remaining methods. A lower n value refers to a stronger shear-thinning behavior [54]. However, HPS-extracted fiber showed an n value of 0.900, indicating a behavior closer to Newtonian fluid characteristics. These results reflected distinct apparent viscosity changes and flow behavior of barley leaf fibers from different extraction methods, further highlighting the importance of polysaccharide extraction methods on their possible applications in diverse food systems.

## 4. Conclusions

In this study, the extraction of different polysaccharides from hulless barley young leaves was investigated using six distinct methods. The resulting water-soluble polysaccharides exhibited distinct compositional profiles. Enzymatic extractions (XE, CE, and XCE) yielded polysaccharides rich in arabinose and galactose, implying a pectic backbone structure, while polysaccharides from AE were abundant in xylose units, revealing an arabinoxylan backbone. HW yielded a polysaccharide enriched in glucose residues, whereas HPS polysaccharides displayed a relatively balanced distribution of glucose, arabinose, galactose, and xylose. Among these methods, the combined enzymatic extraction (XCE) achieved the highest yield (22.63%), while the other methods yielded reasonable amounts (~20%), except for HW extraction (4.11%). The extracted crude polysaccharides exhibited varying molecular weights, ranging from 5.919 × 10^4^ Da to 3.773 × 10^5^ Da across different extraction methods. The FTIR spectra indicated that all six extracted fibers conformed to the general carbohydrate pattern, with some peak variations observed between alkaline extraction and enzymatic extraction methods. Divergent antioxidant activities were observed among the polysaccharides. Alkaline extraction yielded the highest value in the ABTS assay, while enzymatically extracted polysaccharides, despite their higher yield, exhibited lower capacity than those obtained through hot water extraction. An interesting positive correlation was identified between antioxidant capacity and xylose content in the polysaccharide composition, while a negative correlation was observed between antioxidant capacity and galactose and rhamnose content. Significant variations in apparent viscosity were observed among the six fiber samples, with the enzymatically extracted sample displaying a much stronger shear-thinning behavior and higher initial viscosity. Overall, this study contributes to a comprehensive understanding of fiber extraction methodologies from hulless barley leaves, unraveling insights into the compositional and functional attributes of the extracted polysaccharides and offering diverse potential applications in the field of food science.

## Figures and Tables

**Figure 1 foods-12-03381-f001:**
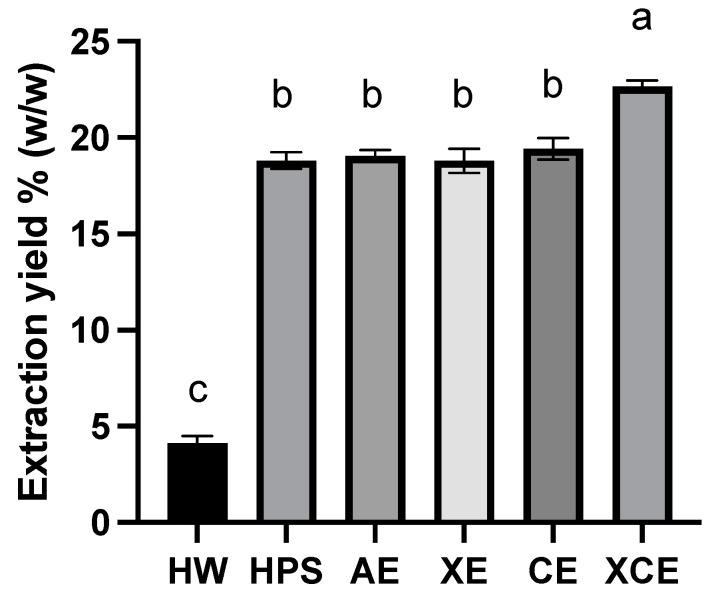
Comparison of extraction yield values from six different extraction methods. Error bars represent the standard deviation calculated from triplicate measurements. Different letters assigned to each bar indicate significant differences among groups (*p* < 0.05). Abbreviations: HW—Hot Water; HPS—High-Pressure Steam; AE—Alkaline Extraction; XE—Xylanase Extraction; CE—Cellulase Extraction; XCE—Combined Xylanase and Cellulase Extraction.

**Figure 2 foods-12-03381-f002:**
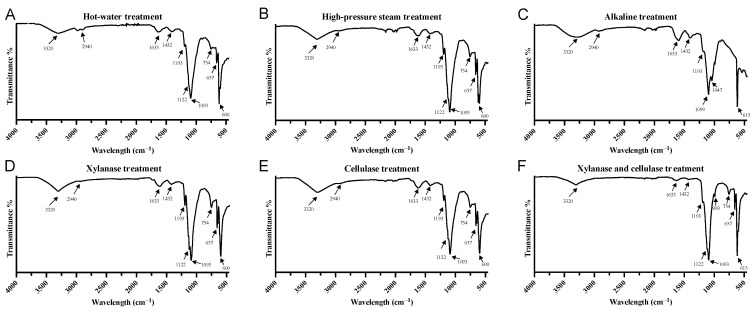
Fourier Transform Infrared (FTIR) spectra for hulless barley leaf fibers extracted using different methods of (**A**) hot water, (**B**) high-pressure stream, (**C**) alkaline, (**D**) xylanase, (**E**) cellulase, and (**F**) combined xylanase and cellulase.

**Figure 3 foods-12-03381-f003:**
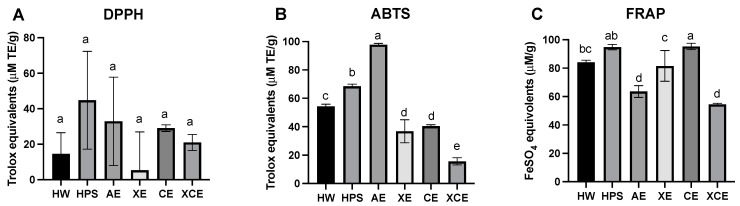
Different antioxidant capacities for hulless barley leaf polysaccharides extracted via six methods. (**A**) DPPH (1,1-diphenyl-2-picryl-hydrazyl) assay; (**B**) ABTS (2,2′-azino-bis(3-ethylbenzothiazoline-6-sulfonic acid)) assay; (**C**) (FRAP) ferric reducing antioxidant power assay. Error bars represent the standard deviation calculated from triplicate measurements. Different letters assigned to each bar indicate significant differences among groups (*p* < 0.05). Abbreviations: HW—Hot Water; HPS—High-Pressure Steam; AE—Alkaline Extraction; XE—Xylanase Extraction; CE—Cellulase Extraction; XCE—Combined Xylanase and Cellulase Extraction.

**Figure 4 foods-12-03381-f004:**
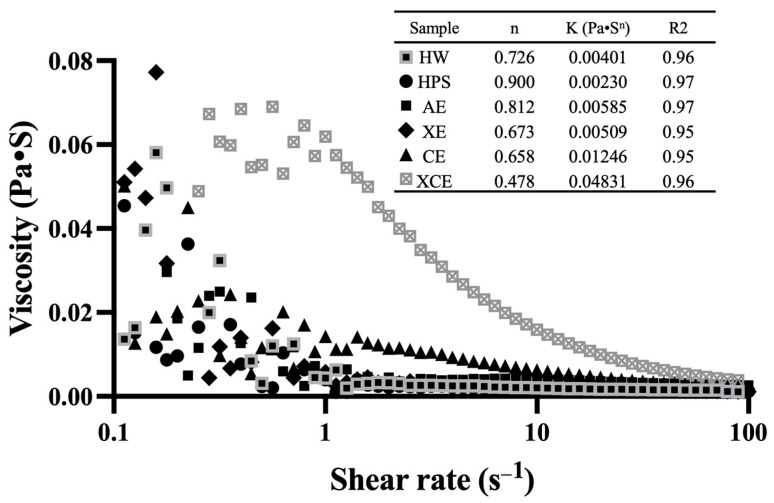
Apparent viscosity and shear flow behavior of hulless barley leaf polysaccharides extracted via different methods. In the inset, K and n denote the power-law model’s flow consistency and flow behavior index, respectively. Abbreviations: HW—Hot Water; HPS—High-Pressure Steam; AE—Alkaline Extraction; XE—Xylanase Extraction; CE—Cellulase Extraction; XCE—Combined Xylanase and Cellulase Extraction.

**Table 1 foods-12-03381-t001:** Monosaccharide composition (mol%) of fibers extracted via six methods ^A^.

Mole Ratio (%)	HW	HPS	AE	XE	CE	XCE
Fucose	0.00 ± 0.00 b	0.57 ± 0.10 a	0.49 ± 0.07 a	0.62 ± 0.13 a	0.00 ± 0.00 b	0.00 ± 0.00 b
Rhamnose	2.57 ± 0.11 d	4.62 ± 0.32 c	2.14 ± 0.19 d	6.82 ± 0.14 b	7.77 ± 1.91 ab	9.24 ± 1.09 a
Arabinose	16.20 ± 0.90 b	18.49 ± 1.81 ab	17.91 ± 0.80 ab	19.02 ± 0.36 ab	18.01 ± 3.76 ab	20.94 ± 0.95 a
Galactose	31.00 ± 1.19 bc	35.05 ± 4.38 bc	24.54 ± 0.84 c	39.38 ± 5.94 ab	47.84 ± 0.71 a	46.34 ± 9.70 a
Glucose	34.00 ± 0.70 a	13.70 ± 3.24 b	13.73 ± 2.05 b	7.06 ± 4.48 b	13.58 ± 12.41 b	9.65 ± 5.26 b
Xylose	11.50 ± 0.40 c	20.57 ± 0.03 b	36.24 ± 2.29 a	13.45 ± 0.21 c	6.17 ± 2.00 d	6.60 ± 0.27 d
Mannose	0.00 ± 0.00 b	0.01 ± 0.01 b	0.16 ± 0.22 b	2.56 ± 0.74 a	0.23 ± 0.16 b	0.00 ± 0.00 b
GalA	3.56 ± 0.93 b	5.12 ± 3.35 ab	3.65 ± 1.58 b	9.30 ± 2.02 a	4.43 ± 3.73 ab	5.05 ± 2.65 ab
GlcA	1.16 ± 0.15 b	1.87 ± 0.03 a	1.15 ± 0.14 b	1.78 ± 0.41 a	1.98 ± 0.16 a	2.18 ± 0.51 a

^A^ Different letter in each row denotes significant differences (*p* < 0.05) among samples.

**Table 2 foods-12-03381-t002:** Molecular patterns of hulless barley young leaf polysaccharides from six extraction methods ^A^.

Sample	Mw (Da)	Mn (Da)	PDI
HW	7.035 × 10^4^ (±1.531%)	6.449 × 10^4^ (±1.622%)	1.091
HPS	5.919 × 10^4^ (±1.641%)	5.122 × 10^4^ (±1.605%)	1.156
AE	6.987 × 10^4^ (±14.466%)	5.905 × 10^4^ (±15.736%)	1.183
XE	3.773 × 10^5^ (±1.323%)	3.332 × 10^5^ (±1.598%)	1.132
CE	2.842 × 10^5^ (±1.201%)	2.708 × 10^5^ (±1.332%)	1.050
XCE	1.441 × 10^5^ (±2.071%)	8.164 × 10^4^ (±3.152%)	1.765

^A^ Abbreviations: Mw—weight average molecular weight; Mn—number average molecular weight; PDI—Polydispersity index. Percentage values in parentheses represent the standard error estimation calculated by MALLS detector.

**Table 3 foods-12-03381-t003:** Pearson coefficient of three antioxidant activity assays and compositions of polysaccharides from six extraction methods.

	DPPH	ABTS ^A^	FRAP
Fucose	0.16	0.5	0.12
Rhamnose	−0.22	−0.87 *	−0.12
Arabinose	−0.03	−0.56	−0.54
Galactose	−0.18	−0.87 *	0.11
Glucose	−0.09	0.22	0.23
Xylose	0.43	0.92 **	−0.21
Mannose	−0.66	−0.24	0.1
GalA	−0.54	−0.41	0.09
GlcA	0.07	−0.74	0.04

^A^ The symbol “*” denotes a significant correlation with *p* < 0.05; “**” indicates *p* < 0.01.

## Data Availability

Data are contained within the article.

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
