# Peer review of "Impact of Six Extraction Methods on Molecular Composition and Antioxidant Activity of Polysaccharides from Young Hulless Barley Leaves"

_foods, 2023, doi:10.3390/foods12183381_

Round 1
Reviewer 1 Report
The presented manuscript “Impact of Six Extraction Methods on Molecular Composition and Antioxidant Activity of Polysaccharides from Young Hulless Barley Leaves” presents a good exercise of chemical analysis of extracts obtained by different methods. The manuscript is correctly written, and the methodology was correctly applied. The following is in order to improve the manuscript.
Line 113. Remove the double .
Line 182. Remove the double space.
Line 386. Check the activity reported, change antibiotic capacity for antioxidant capacity.
My major concerns are in the methodology section.
Please, explain the reason for use 80°C and 2 hours for hot water extraction. Also, the solvent:solid ratio 25/1. In my understanding, these conditions were applied in all extraction processes. Therefore, it is important to define what is the reason for these parameters.
In this sense, according with lines 236 – 239, the hydroxyl ions concentration has a significative effect on extraction process, that means that the concentration of NaOH is important, if the authors increase the concentration, the yield improve?
Reviewer 2 Report
Hulless barley leaves have garnered increasing attention due to their potential health benefits. It's crucial to carefully choose the method for extracting polysaccharides from these leaves, as this not only impacts their functional properties but also their potential applications in the food industry. The authors of this study have conducted a comprehensive comparative analysis of six different fiber extraction techniques: hot water extraction, high-pressure steam extraction, alkaline extraction, xylanase extraction, cellulase extraction, and a combination of xylanase and cellulase extraction. The analysis encompasses a detailed examination of several key aspects among fibers extracted through these methods, including monosaccharide composition, molecular weight, and antioxidant activities assessed through DPPH, ABTS, and FRAP assays. The findings reveal essential insights into the advantages and limitations of each extraction method.
In terms of antioxidant activities, alkaline extraction produced the highest value in the ABTS assay, suggesting strong antioxidant potential. Interestingly, despite the higher yield achieved through enzymatic methods, the enzymatically extracted polysaccharides exhibited lower antioxidant capacity. This observation suggests that the extraction method plays a crucial role in influencing the functional properties of the extracted fibers.
In summary, this study demonstrated the significance of selecting an appropriate extraction method for obtaining polysaccharides from hulless barley leaves. The choice of method can significantly impact the yield, monosaccharide composition, molecular weight, and antioxidant activities of the extracted fibers, ultimately influencing their suitability for various applications in the food industry and beyond.
The research is well structured and executed. Nevertheless, there are some recommendations and remarks that need to be incorporated.
· The mechanism of antioxidant activity is not very clear. How and why do fibers, which are polysaccharides, exhibit antioxidant properties?
· FTIR analysis of the fibers must be performed. An in-depth analysis of the spectral analysis is required.
· The viscosity-improving capacity of the fibers should be analyzed by viscometry/ rheometry.
The quality of English in the given text is generally good. The text is clear and well-structured, and it effectively conveys the research findings. However, there are a few minor scopes for improvement, which authors should do by careful proofreading.
Reviewer 3 Report
The manuscript "Extraction Methods on Molecular Composition and Antioxidant Activity of Polysaccharides from Young Hulless Barley Leaves" is very interesting.
I have a small number of minor comments that I believe that authors should address.
Ln 31. Please present the names of biological species in italics according to traditional nomenclature (Hordeum vulgare L. var. nudum Hook. f).
The sentence staring at Ln 55 makes little sense in its current structure. The way the sentence is structured implies that the text following the word "and" relates to the polysaccharides. Antioxidants are not polysaccharides.
Ln 92, I assume the grain were "soaked" and not "socked"
Figure 1. Please supply a caption that explains the letters above the bars. Also provide an explanation of what the error bars mean. Reading from left to right, it would make sense that you follow the alphabet in your superscripts (a, bc, d). Are you sure that the superscript for HW is "c" and not "d"? You are reporting the "extraction yield", but what is that extraction yield. Is that dry matter, is that polysaccharides?????? Also, in the caption, explain the abbreviations
Table 1. Are you trying to say that galactose is the dominant monosaccharide in barley leaves? This sound extremely high.
Table 2. please provide a detailed caption that explains: "Mn", PDI, and the values in brackets.
Figure 2. Why are you using colours in this figure but not in Figure 1?
